# Modeling the 2014–2015 Vesicular Stomatitis Outbreak in the United States Using an SEIR-SEI Approach

**DOI:** 10.3390/v16081315

**Published:** 2024-08-18

**Authors:** John M. Humphreys, Angela M. Pelzel-McCluskey, Phillip T. Shults, Lauro Velazquez-Salinas, Miranda R. Bertram, Bethany L. McGregor, Lee W. Cohnstaedt, Dustin A. Swanson, Stacey L. P. Scroggs, Chad Fautt, Amber Mooney, Debra P. C. Peters, Luis L. Rodriguez

**Affiliations:** 1Foreign Animal Disease Research Unit, Agricultural Research Service, U.S. Department of Agriculture, Plum Island Animal Disease Center (PIADC) and National Bio Agro Defense Facility (NBAF), Manhattan, KS 66502, USA; lauro.velazquez@usda.gov (L.V.-S.); miranda.bertram@usda.gov (M.R.B.); chad.fautt@usda.gov (C.F.); amber.mooney@usda.gov (A.M.); luis.rodriguez@usda.gov (L.L.R.); 2Veterinary Services, Animal and Plant Health Inspection Service (APHIS), U.S. Department of Agriculture, Fort Collins, CO 80526, USA; angela.m.pelzel-mccluskey@usda.gov; 3Arthropod-Borne Animal Disease Research Unit, Agricultural Research Service, U.S. Department of Agriculture, Manhattan, KS 66502, USA; phillip.shults@usda.gov (P.T.S.); bethany.mcgregor@usda.gov (B.L.M.); stacey.scroggs@usda.gov (S.L.P.S.); 4Foreign Arthropod-Borne Animal Diseases Research Unit National Bio- and Agro-Defense Facility, Agricultural Research Service, U.S. Department of Agriculture, Manhattan, KS 66502, USA; lee.cohnstaedt@usda.gov; 5Center for Grain and Animal Health Research, Agricultural Research Service, U.S. Department of Agriculture, Manhattan, KS 66502, USA; dustin.swanson@usda.gov; 6Oak Ridge Institute for Science and Education (ORISE)-NBAF, Oak Ridge, TN 37831, USA; 7Office of National Programs, Agricultural Research Service, U.S. Department of Agriculture, Beltsville, MD 20705, USA; deb.peters@usda.gov

**Keywords:** vesicular stomatitis, vector-borne diseases, epidemiology, transmission, compartmental model

## Abstract

Vesicular stomatitis (VS) is a vector-borne livestock disease caused by the vesicular stomatitis New Jersey virus (VSNJV). This study presents the first application of an SEIR-SEI compartmental model to analyze VSNJV transmission dynamics. Focusing on the 2014–2015 outbreak in the United States, the model integrates vertebrate hosts and insect vector demographics while accounting for heterogeneous competency within the populations and observation bias in documented disease cases. Key epidemiological parameters were estimated using Bayesian inference and Markov chain Monte Carlo (MCMC) methods, including the force of infection, effective reproduction number ($R_{t}$), and incubation periods. The model revealed significant underreporting, with only 10–24% of infections documented, 23% of which presented with clinical symptoms. These findings underscore the importance of including competence and imperfect detection in disease models to depict outbreak dynamics and inform effective control strategies accurately. As a baseline model, this SEIR-SEI implementation is intended to serve as a foundation for future refinements and expansions to improve our understanding of VS dynamics. Enhanced surveillance and targeted interventions are recommended to manage future VS outbreaks.

## 1. Introduction

Vesicular stomatitis (VS) is a livestock disease caused by the vector-borne vesicular stomatitis New Jersey virus (VSNJV) or vesicular stomatitis Indiana virus (VSIV), both of which belong to the Rhabdoviridae family [1]. Over the past two decades, VSNJV has been the predominant serotype responsible for outbreaks in the United States (U.S.), including the 2014–2015 outbreak [2]. This study aims to enhance the mechanistic understanding of VSNJV transmission dynamics and elucidate the underlying mechanisms that drove the 2014–2015 VS outbreak. The 2014–2015 VS outbreak was first detected in Kinney County, Texas, in May 2014, and continued through the final release of all quarantined premises on 4 March 2016. A total of 3173 animals (2095 equids and 1078 bovids) were confirmed to be infected by VSNJV during the outbreak, necessitating quarantine of 1258 individual premises [3]. Given the clinical similarity of VS symptoms in cattle and swine to those of foot-and-mouth disease (FMD), a severe livestock disease eradicated from the U.S. in the 1920s [1], the detection of VSNJV-type lesions in the U.S. is subject to mandatory reporting to rule out FMD [2]. VS outbreaks in the U.S. significantly disrupt the equine, cattle, and swine industries through quarantines that restrict animal movement to reduce disease spread. These quarantine measures lead to industry-wide economic losses caused by international trade restrictions, interstate movement restrictions, additional veterinary expenses, and business interruptions from canceled shows, fairs, and other livestock events [4,5,6,7].

Equids and cattle are most frequently affected by VSNJV in the U.S.; however, domestic swine and captive non-domestic species can also be infected [2,8]. VSNJV infection typically presents with vesicular lesions on the muzzle, tongue, lips, nostrils, oral mucosa, teats, sheath, and coronary bands. These blister lesions can lead to excessive salivation and a loss of appetite, particularly when located around the muzzle or in the oral cavity, or result in lameness when found on the coronary bands [2,6]. Although viremia has not been found in infected vertebrate hosts, the primary mechanism of VSNJV transmission is through biting, hematophagous insect vectors that become infected through blood feeding on or near lesioned skin of affected livestock, or by the vector-to-vector exchange on non-viremic hosts (co-feeding) [9,10,11,12]. However, the virus can be spread via contact with infectious lesions and saliva or indirectly through contaminated water, feed, veterinary equipment, or other fomites [13,14]. Suspected VSNJV vectors include black flies (Simuliidae), sand flies (Psychodidae), and *Culicoides* midges (Ceratopogonidae), although, other biting insects like mosquitoes may also serve as mechanical VSNJV vectors [15,16,17].

As outlined by the incursion-expansion hypothesis [18,19,20,21], the pattern of VS outbreaks observed in the U.S. typically follows a decadal cycle marked by distinct phases of incursion and expansion. According to this proposed framework, each cycle begins with the springtime introduction of VSNJV from endemic regions in southern Mexico into the southwestern U.S., followed by overwintering, subsequent spread, and local extinction within one to three years [22,23,24]. These periodic outbreaks, alternating between active years and intervals of eight to ten years of dormancy, often expand northward from border states such as Texas, New Mexico, and Arizona to northern regions including Colorado, Utah, Wyoming, and Nebraska [7,21]. Given VS’s disruption and economic impact on the U.S. livestock industry, an improved understanding of the disease’s transmission process is needed to develop effective control and prevention strategies.

Compartmental models are mechanistic models that employ mathematics to simulate biological or physical processes through a system of ordinary differential equations [25]. SEIR models are a class of compartmental models applied in epidemiology that aid in understanding infectious disease by dividing a host population into discrete compartments reflecting individuals susceptible (S) to a pathogen, exposed (E) to the pathogen but not yet infectious to others, infectious (I) to other members of the population, or removed (R) from the transmission chain due to quarantine, recovery, or death [26,27,28]. Initially conceptualized to explore theoretical ideas such as basic reproduction numbers and immunity thresholds within host populations, compartmental models are increasingly applied to transmission questions in arboviral diseases [29,30,31,32,33].

Compartmental models like the SEIR have not been applied to VS previously; however, the models may be valuable in estimating metrics that are crucial for managing VS, such as the basic reproduction number ($R_{0}$) and the force of infection (FOI), as well as quantifying uncertainty around what is currently understood about VSNJV-specific mechanisms like the extrinsic incubation period (EIP), the intrinsic incubation period (IIP), recovery rates, and vector-host transmission probabilities. The EIP is the time interval between when an arthropod vector acquires a pathogen through feeding and when it can transmit the pathogen to a new host. In contrast, the IIP is the lag between when a vertebrate host is infected with a pathogen and when the host becomes infectious to others in the transmission chain. This ability to quantify these mechanisms makes compartmental models essential for approximating transmission rates from reported cases and evaluating the effectiveness of proposed interventions and response strategies [33].

This study introduces an SEIR-SEI compartmental model tailored for VS transmission processes in the U.S. The model is based on the pioneering Ross–MacDonald methodology [26,27,28] that combines arthropod vector and vertebrate host dynamics in a common model. That is, in addition to segmenting the host population into S, E, I, and R compartments, the model also divides the vector population into susceptible (S), exposed (E), and infectious (I) arthropods that interact with hosts as an integrated system. Building on this foundation, our VS model incorporates elements that distinguish it from otherwise similar arthropod-borne vector models applied to human diseases such as malaria [32,34], Zika [30,35], dengue [36,37], and West Nile [31,33], including model parameters to help account for observation bias and heterogeneous competence in the host and vector populations.

In an epidemiology context, competence refers to an organism’s capacity to transmit a pathogen following exposure and is a critically important attribute to include in modeling disease dynamics [38,39]. Although competence can be considered to operate across all levels of biological organization, from the genotype to individuals, populations, and entire species, we applied a more narrow framework to quantify heterogeneous competency among VSNJV vectors and hosts.

In the case of vectors, we use heterogeneous competency ($\rho_{v}$) to refer to an arthropod’s intrinsic capacity to transmit a pathogen after exposure. This encompasses the pathogen’s ability to enter and replicate within the vector, spread to the vector’s salivary glands, and be released into the saliva at a sufficiently high concentration to infect the next vertebrate host in the transmission chain, or to be passed to other vectors through co-feeding. Because the SEIR-SEI model is implemented at the population level, vector heterogeneous competency quantifies the proportion of the exposed vector population that becomes infectious despite factors such as the vector’s immune response, constraining pathogen–microbiome interactions, decreased viral or vector fitness, and physiological mechanisms like the midgut barrier [24,40] that otherwise limit or prevent further transmission.

Like vector competence, host competence varies due to genetic, physiological, and behavioral characteristics mediating post-exposure infectiousness. A dominant physiological determinant of host competence in the VS system is lesion presentation. Lesions connected to clinical VS increase host competence and transmission rates by introducing infectious material that facilitates insect feeding. This promotes animal-to-animal transmission through direct contact and enables indirect transmission by shedding virus-laden material into the environment [6]. Livestock showing severe disease likely spread VSNJV more rapidly and over greater distances than animals with mild or unapparent signs [41]. As applied to vertebrate hosts, we use heterogeneous competency ($\rho_{h}$) to distinguish symptomatic from non-symptomatic infections and to enumerate the proportion of the exposed host population that disproportionately contributes to VSNJV transmission through clinical disease presentation.

In addition to heterogeneous competency, the proposed model addresses the issue of biased, imperfect disease detection by incorporating a parameter (κ) that simulates reported disease cases and the total number of undetected and unreported cases. This observation bias term aids in more accurately depicting an outbreak’s impact when surveillance or under-reporting results in underestimating case numbers, a common problem in many arboviral systems [42,43,44,45]. In addition to the impossibility of perfect surveillance across the entirety of the Western U.S., anecdotal evidence from state and federal animal health officials suggests that VS cases are under-reported or underestimated during outbreaks. For instance, livestock owners have admitted to not reporting suspected cases during post-outbreak meetings due to fatigue and familiarity with the disease. In response to VS detection, these individuals indicated that affected animals were isolated but not documented with authorities. This pattern of non-reporting may be more prevalent in historically affected areas, where owners recognize the disease and avoid the burden of quarantines, as well as in newly affected states where livestock owners may lack awareness about the importance of reporting symptoms. Anecdotal evidence such as this underscores the necessity of model bias correction to represent the outbreak’s true extent accurately.

We implemented a Bayesian modeling workflow [46,47] using the Stan programming language [48] and Markov chain Monte Carlo (MCMC) methods to provide robust parameter estimation, uncertainty assessment, and integration of VSNJV prior knowledge. By incorporating heterogeneous competency within host and vector populations and capturing the under-reported dimension of disease spread, this study aims to enhance mechanistic understanding of VSNJV transmission dynamics and elucidate the underlying mechanisms that drove the 2014-2015 VS outbreaks. Furthermore, an interactive dashboard and the code needed to construct the model are provided to explore result implications, reproduce the analysis, or adapt our approach to other disease systems (see Data Availability Statement).

## 2. Data and Methods

### 2.1. Study Domain and Observed Incidence

This study used VS disease observation data documented by Federal, state, county, and local animal health professionals during the 2014–2015 outbreak in the U.S. The outbreak encompassed a geographic area of approximately 2.5 million km^2^ in the Western U.S., including premises in the states of Texas (TX), New Mexico (NM), Arizona (AZ), Colorado (CO), Utah (UT), Nebraska (NE), Wyoming (WY), and South Dakota (SD).

The U.S. Department of Agriculture (USDA) Animal and Plant Health Inspection Service (APHIS), Veterinary Services provided VS disease data [3], which included 3173 veterinarian-confirmed detections of VS disease documented from 2095 equids and 1078 bovids located on 1258 unique premises (private residences, ranches, farms, businesses, or other facility locations). In 2014, confirmation of VS necessitated the animals to exhibit both (1) clinical disease (lesions) and (2) have VSNJV infection confirmed through laboratory or diagnostic testing. In 2015, VS confirmation required the (1) presentation of lesions and (2) the suspected case to be located in a county previously confirmed through laboratory or diagnostic testing to have VS occurrence. The recorded incidence included the estimated date of disease onset based on animal examination and owner interviews. In addition to observed incidence, the estimated number of susceptible animals co-located within the same herd or at the same premises as confirmed VS cases was documented as 13,513 in 2014 and 21,846 in 2015. The figures depicting observed incidence are combined with other information in the results section.

### 2.2. Mathematical Model

We developed an SEIR-SEI mathematical model to describe VSNJV transmission dynamics among vertebrate hosts and insect vector populations during the 2014–2015 VS outbreak in the U.S. The model divides the total vertebrate host population ($N_{h}$) into four compartments; susceptible ($S_{h}$), exposed ($E_{h}$), infectious ($I_{h}$), and removed ($R_{h}$) in which $N_{h}=S_{h}+E_{h}+I_{h}+R_{h}$; see Figure 1. For the vector population ($N_{v}$), a susceptible-exposed-infectious (SEI) structure was adopted comprising susceptible ($S_{v}$), exposed ($E_{v}$), and infectious ($I_{v}$) vectors with $N_{v}=S_{v}+E_{v}+I_{v}$. Due to their short lifespan, infected vectors were assumed not to recover; therefore, a removal or recovery compartment was unnecessary.

As a compartmental model, homogeneous mixing was assumed within the population, indicating random contact among individuals without preference for smaller subgroups or specific geographic locations of elevated risk. Although a limited number of re-infections in the same animal is possible [2], the model assumes this rate as negligible, with recovered hosts remaining immune for the outbreak year. These assumptions prioritize rapid epidemic dynamics over host demographic changes like births and natural deaths, which are considered negligible due to the short timeframe and non-endemic dynamics [18,19,20] of VS outbreaks in the U.S.

The number of individual hosts or vectors within each model compartment varies by time (daily) as determined by rate parameters reflecting the current understanding of VSNJV transmission dynamics and epidemiology (Figure 1, Table 1). Because compartmental models have not previously been applied to analyze VS outbreaks, literature reporting laboratory and experimental VS studies were used to identify reasonable parameter ranges [24,40,53,58]. In addition to VSNJV-specific research, publications that utilized mathematical modeling to assess African horse sickness [51,59], bluetongue virus disease [49,54,55,60,61], and epizootic hemorrhagic disease [62] were also referenced in hope of identifying parameters for vector groups held in common with VS (e.g., Simuliidae, Psychodidae, and Ceratopogonidae). Ultimately, however, most available parameters reflected *Culicoides* midge characteristics; therefore, the results are interpreted for that vector group. Parameter ranges detailed in Table 1 were evaluated through iterative fitting, parameter optimization, and comparison to APHIS documented incidence to identify the best-performing values. Due to the assumption of sustained, post-recovery immunity by hosts within an outbreak year, data from 2014 and 2015 were separately analyzed. This approach also helped resolve numeric issues resulting from abundant zeros (no observed VS cases) during winter and facilitated between-year comparisons.

The rate at which the members of the susceptible host population ($S_{h}$) become exposed ($E_{h}$) is governed by the force of infection (FOI,$\lambda_{h}$). The FOI is a function of the proportion of infectious vectors in the environment ($\frac{I_{v}}{N_{v}}$), the contact rate ($\upsilon_{h}$) with those infectious vectors, and the probability of virus transmission ($\beta_{h}$) given a successful vector bite (Equation (Equation 1)). The product of the FOI and $S_{h}$ determines the exposed host number (Equation (2)), which, following a latent period of within-host incubation ($\sigma_{h}$, IIP), competent hosts ($\rho_{h}$) progress to become infectious ($I_{h}$, Equation (4)) and potentially transmit VSNJV to other hosts through direct and indirect contact or to new vectors during vector feeding ($\lambda_{v}$). Because observed incidence data reflects hosts with clinical symptoms, competency ($\rho_{h}$) approximates the proportion of simulated exposed hosts that would likewise show symptoms. Infectious hosts are removed from the system at rate $\gamma_{h}$ and assumed to be immune for the remainder of the outbreak cycle.
(1)$$\begin{array}{cc} \lambda_{h} & =\upsilon_{h}\cdot \beta_{h}\cdot \frac{I_{v}}{N_{v}} \end{array}$$
(2)$$\begin{array}{cc} \frac{dS_{h}}{dt} & =-\lambda_{h}\cdot S_{h} \end{array}$$
(3)$$\begin{array}{cc} \frac{dE_{h}}{dt} & =\lambda_{h}\cdot S_{h}-\sigma_{h}\cdot E_{h} \end{array}$$
(4)$$\begin{array}{cc} \frac{dI_{h}}{dt} & =\sigma_{h}\cdot E_{h}\cdot \rho_{h}-\gamma_{h}\cdot I_{h} \end{array}$$
(5)$$\begin{array}{cc} \frac{dR_{h}}{dt} & =\gamma_{h}\cdot I_{h} \end{array}$$

Susceptible members of the vector population ($S_{v}$) are exposed to VSNJV at a rate that is also governed by an FOI function (Equation (Equation 6)), but an FOI dependent on available infectious hosts ($\frac{I_{h}}{N_{h}}$), the number of bites delivered to those hosts ($\upsilon_{v}$), and the probability of VSNJV uptake ($\beta_{v}$) for each of those bites. A background mortality rate (μ) affects all vector compartments and is balanced by an assumed equivalent birth rate. As with hosts, exposed vectors ($E_{v}$) experience a latent period ($\sigma_{v}$, EIP) before a subset of competent individuals ($\rho_{v}$) become infectious ($I_{v}$, Equation (8)). Equation (9) shows that the number of infectious vectors includes only the competent proportion of the population that achieves post-exposure infectivity and can transmit the virus (heterogeneous competency, $\rho_{v}$).
(6)$$\begin{array}{cc} \lambda_{v} & =\upsilon_{v}\cdot \beta_{v}\cdot \frac{I_{h}}{N_{h}} \end{array}$$
(7)$$\begin{array}{cc} \frac{dS_{v}}{dt} & =\mu_{v}\cdot N_{v}-\lambda_{v}\cdot S_{v}-\mu_{v}\cdot S_{v} \end{array}$$
(8)$$\begin{array}{cc} \frac{dE_{v}}{dt} & =\lambda_{v}\cdot S_{v}-\left(\sigma_{v}+\mu_{v}\right)\cdot E_{v} \end{array}$$
(9)$$\begin{array}{cc} \frac{dI_{v}}{dt} & =\sigma_{v}\cdot E_{v}\cdot \rho_{v}-\mu_{v}\cdot I_{v} \end{array}$$

Although SEIR-SEI outputs can be interpreted as predictions, the approximations produced by the model vary more smoothly than the discrete and often noisy incidence data collected during surveillance [63]. Because of this, a negative binomial distribution was used to reconcile the differences between model predictions and the observed case data. Transitions between compartments can be modeled either deterministically or stochastically. The present model is deterministic because it numerically solves ordinary differential equations (ODEs) to produce consistent outcomes for a given set of parameters and initial conditions. In contrast, stochastic models incorporate random variation in compartment transitions and are typically constructed using stochastic differential equations (SDEs), discrete event simulations, or agent-based models. Although our model is deterministic, we incorporate a sampling distribution to account for uncertainty in estimates derived from imperfect surveillance data and an incomplete understanding of VSV transmission mechanisms. The negative binomial distribution is commonly applied in epidemiological analysis to allow for overdispersion [64] and is specified below with prior assumptions,
(10)$$\begin{array}{cc} Y_{t}|\theta & ∼\mathrm{NegBinomial}\;(Y|\Delta I_{t}\kappa ,\phi )\\ \theta & =\{\upsilon_{h},\upsilon_{v},\beta_{h},\beta_{v},\sigma_{h},\sigma_{v},\gamma_{h},\rho_{h},\rho_{v},\mu_{v},\kappa \}\\ \upsilon_{h} & ∼\mathrm{Lognormal}\;(log(0.25),0.1)\\ \upsilon_{v} & ∼\mathrm{Lognormal}\;(log(0.15),0.1)\\ \beta_{h} & ∼\mathrm{Beta}\;(10,1)\\ \beta_{v} & ∼\mathrm{Beta}\;(1,10)\\ \sigma_{h} & ∼\mathrm{Lognormal}\;(log(0.17),0.1)\\ \sigma_{v} & ∼\mathrm{Lognormal}\;(log(0.3),0.1)\\ \gamma_{h} & ∼\mathrm{Lognormal}\;(log(0.15),0.1)\\ \rho_{h} & ∼\mathrm{Beta}\;(10,100)\\ \rho_{v} & ∼\mathrm{Beta}\;(5,10)\\ \mu_{v} & ∼\mathrm{Lognormal}\;(log(0.1),0.1)\\ \kappa & ∼\mathrm{Beta}\;(15,100)\\ 1/\phi & ∼\mathrm{Exponential}\;(2) \end{array}$$
where *Y* is the observed VS incidence at time *t* and is conditional on θ, the set of parameters encoding what is believed about VSNJV transmission dynamics. The expected value for the negative binomial is estimated from the infectious compartment ($I_{h}$); however, because the infectious compartment gives prevalence (total number of infectious individuals on day *t*), $\Delta I_{t}$ is used to estimate incidence (only the new cases on day *t*). The expected value is then scaled by κ to account for observation bias and under-reporting. Note that the overdispersion parameter ϕ is reparameterized using its inverse 1/ϕ to prevent the prior distribution from overly favoring models that assume too much overdispersion [47]. The prior for 1/ϕ is specified as an exponential distribution with rate parameter 2, which encodes belief about how much variation is expected before observing the data.

The Results and Discussion section provides the biological justification and interpretation of θ parameters. However, the informative prior distributions shown in Equation (Equation 10) demonstrated the best performance in estimating VS incidence following sensitivity analyses. To avoid boundary issues caused by estimated parameters reaching the limits of parameter space, log-normal distributions were chosen for parameters believed to be positive real numbers, and beta distributions were selected for parameters in the censored 0–1 range used for probabilities, percentages, and proportions.

Bayesian models were developed in Stan and utilized the Runge-Kutta (RK45) solver for non-stiff ODE systems [48,65,66], which enabled parameter inference and simulation-based prediction [47]. This flexibility is advantageous in epidemiological modeling, where the framework can quantify the uncertainty surrounding VSNJV transmission parameters. We utilized Bayesian inference methods, specifically MCMC with a Hamiltonian Monte Carlo (HMC) sampler [67], to provide a principled uncertainty assessment. Following a 500-iteration burn-in phase, four independent chains of 2500 iterations were executed with initial values for each chain randomly drawn from prior distributions. This approach allows us to integrate existing knowledge about VSNJV transmission through the use of informative prior distributions.

Model evaluation and diagnostics encompassed checks for MCMC convergence and the effective sample size (ESS) as presented in work by Stan [48,65,66], along with graphical and numerical comparisons of prior and posterior predictive plots against observed incidence data using the bayesplot R-package [46]. Leave-one-out cross-validation employing Pareto smoothed importance sampling (PSIS-LOO) was also conducted [68]. Sensitivity analyses were performed for all initial values, the total host population size ($N_{h}$), the total vector population size ($N_{v}$), and the date of virus introduction. Modifications to prior distributions for other model parameters listed in Table 1 were evaluated for sensitivity across a 0.05 to 1.00 standard deviation range.

## 3. Results and Discussion

This study emphasized compartmental models as tools for elucidating VSNJV transmission dynamics and offered a framework for assessing the 2014–2015 VS outbreak in the U.S. Integrating vector and host populations within the SEIR-SEI model improved the mechanistic understanding of VSNJV epidemiology, which is essential for developing effective control strategies. Our model’s ability to incorporate heterogeneous competency within vectors and hosts, and to correct for observation bias, provided a more accurate depiction of outbreak incidence. This enhancement of the model’s predictive power and potential applicability improves our capacity to respond to future disease response.

Simulated dynamics from the SEIR-SEI model (Figure 2 and Figure 3) indicated that the earliest probable date that the 2014–2015 outbreak cycle began was 3 February 2014, 109 days before the APHIS documented index case of 23 May 2014. Although the estimated start of the outbreak precedes the first documented case by several months, this timing is consistent with VSNJV detections made in Mexico’s Sonora and Chihuahua States in late 2013 and early 2014 [69]. This timing is also supported by genomics analyses and transmission chain reconstructions, which showed a sharp increase in VSNJV diversity and active transmission in mid-January 2014 [70]. Model estimated observation bias for 2014 (Figure 4, κ in Table 1) displayed a median of 0.09 (0.08–0.10 95% CI) suggesting that only 10% of vertebrate host infections were documented during 2014. Heterogeneous competency among hosts ($\rho_{h}$) indicated that 20% (16–26% 95% CI) of all observed and unobserved 2014 cases represented clinical disease. This rate of clinical presentation among infected hosts is comparable to that found for Colorado horses by Urie et al. [6] during the 2014 outbreak. The study by Urie et al. [6] is discussed further below but estimated that 28.6% of serologically confirmed VSNJV infections showed no clinical signs of infection.

Simulated 2015 dynamics (Figure 3) suggested that the outbreak year began as early as 7 March 2015, 53 days before the recorded index case on 29 April 2015. Observation bias for 2015 was slightly improved over 2014 with a median κ of 0.24 (0.20–0.28 95% CI), indicating that about 24% of predicted infections were documented in the second year of the epizootic (Figure 4).

Comparison of VSNJV vector and host prevalence indicated that peak vector prevalence occurred after maximum vertebrate host prevalence and approximately 43 days earlier in the year during 2014 than in 2015 (Figure 5). These findings suggest that delayed vector population dynamics between years and within-year co-occurrence of multiple vector species may influence VS transmission patterns. Environmental factors such as temperature fluctuations, precipitation variability, or physical habitat alterations could drive delayed vector population dynamics, affecting vector development rates and emergence times [71,72]. Additionally, VS transmission is suspected to involve multiple vector species, each with distinct environmental preferences and unique seasonal emergence patterns [17,24,41,73]. Vector and host diversity can decouple spatial and temporal trends in disease systems [74], and species-specific differences could result in asynchronous peaks in the vector populations relative to prevalence changes in the vertebrate host population [75].

Heterogeneous competency among hosts ($\rho_{h}$) indicated that 24% (19–30% 95% CI) of all observed and unobserved 2015 cases represented clinical disease. Like the $\rho_{h}$ estimated for 2014, this rate of clinical presentation is consistent with the Urie et al. [6] study, which showed 28.6% of confirmed VSNJV infections to be non-clinical.

On average, κ parameters for the entire outbreak cycle imply that only about a fifth (17%) of VSNJV-infected vertebrate hosts were documented during the 2014–2015 U.S. epizootic. Recognizing that model-estimated counts reflect the theoretical number of cases necessary for sustaining disease propagation, including observed and unobserved infections, the discrepancy between simulated and observed cases is unsurprising for several reasons.

First, clinical presentation was the diagnostic criterion for establishing disease occurrence in this study, but not all infected hosts become symptomatic. For example, a Urie et al. [6] study investigating VS outbreaks on managed premises in Colorado found that 71.4% of 147 horses confirmed through serological testing to be exposed to VSNJV in 2014 exhibited clinical signs of disease, whereas 28.6% showed no clinical signs of infection. Although clinical hosts almost certainly facilitate disease spread at higher rates, viruses shed in the saliva, nasal secretions, and feces of sub-clinical animals may also contribute to VSNJV propagation [10,41,76].

Second, owners or managers may not recognize mild or inconspicuous infections in domestic livestock, leading to underreporting. Most cases documented in 2014 occurred in horses, generally subject to more frequent and routine owner and veterinary contact than cattle and other potential livestock hosts. A comparison of 2014–2015 clinical cases to the number of susceptible animals during the same period shows that although 24,682 cattle were documented, only 1078 (4.34%) were confirmed to have VS, whereas 15.3% of the 13,850 horses exhibited symptoms. As described in the introduction, anecdotal evidence shows that non-reporting may occur to avoid quarantine, even when symptoms are recognized.

Third, VS may be propagated from uninfected vertebrate hosts through non-conventional mechanisms. Due to the absence of detectable viremia, livestock have traditionally been considered dead-end hosts. However, even uninfected cattle can potentially serve as sources of VSNJV transmission by facilitating horizontal transmission among co-feeding vectors or providing a substrate for uninfected vectors to acquire VSNJV from sites where infected vectors previously fed [12]. The presented SEIR-SEI model did not include co-feeding as a distinct transmission mechanism. Therefore, the model would have included or lumped functional contributions through such non-conventional means with other host interactions.

Fourth, VSNJV may spread in unidentified wildlife reservoirs not included in routine surveillance efforts. VSNJ antibodies have been confirmed in several wildlife species, including deer, elk, pronghorn, bighorn sheep, bears, bobcats, rodents, rabbits, birds, raccoons, and opossums [77,78,79,80]. Although these species are not suspected of playing a major role in U.S. epizootics [79], they evidence the broad range of hosts potentially susceptible to VSNJV infection. Other wildlife, such as invasive or feral swine, show a higher potential for transmitting VSNJV. Swine are natural VSNJV hosts [14,81,82], and feral swine may have contributed to enzootic maintenance on Ossabaw Island, Georgia [57]. Ossabaw Island was the only known endemic focus of VSNJV in the U.S., and the disappearance of VSNJV there was contemporaneous with feral swine depopulation efforts [57]. Feral swine are abundant in southern portions of the U.S. and Texas, which borders Mexico, has more feral swine than any other U.S. state, with a recently estimated population size exceeding 2.5 million animals [83,84]. Wildlife infections may spread the virus along intermittent or stuttering transmission chains below the epidemic thresholds that would enable detection [85,86].

The discrepancy between model simulations and observed cases underscores the importance of considering all potential hosts and transmission pathways in interpreting disease dynamics. By estimating the true burden of infection, the model provides a more comprehensive picture of the outbreak, highlighting the necessity for enhanced surveillance and reporting systems to capture the full extent of disease spread in both domestic and wild populations [87,88]. Without considering potential wildlife hosts [77,78,79,80], livestock numbers and density in the Western U.S. are sufficient to propagate VS. According to 2024 agricultural statistics, there are approximately 87.2 million head of cattle in the U.S., and the top two cattle-producing states are all located in the area affected by the 2014–2015 outbreaks; Texas has the most cattle, at 12.5 million head, followed by Nebraska with 6.3 million [89]. Similarly, a high proportion of the 7.2 million horses in the U.S. are held by Texas (767,000) and Nebraska (180,000) [90]. Model results suggest that VS cases in some of these animals may be undetected or unreported.

The effective reproduction number ($R_{t}$) is an epidemiological metric that indicates the average number of secondary infections generated by a single infected individual at a specific time point during an ongoing outbreak. Unlike the basic reproduction number ($R_{0}$), which assumes a fully susceptible population, $R_{t}$ tracks changes in population size, susceptibility, or other dynamic factors. The $R_{t}$ enables comparison between outbreaks to gauge the relative intensity and spread of the same disease during different times. Comparing the vertebrate host $R_{t}$ across the outbreak cycle shows that VS intensity was similar in 2014 and 2015 (Figure 6). Although 2015 $R_{t}$ remained above 1.0 about a week later into the year than in 2014, it declined similarly during both years, falling below 1.0 around early November, signifying non-epidemic transmission rates.

**Figure 6 viruses-16-01315-f006:**
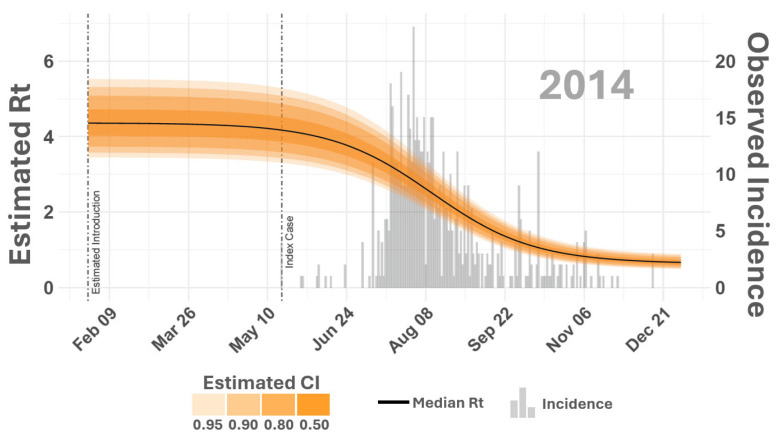
Effective reproduction number ($R_{t}$). The figure displays the estimated effective reproduction number ($R_{t}$) for 2014. The vertical axes on the left correspond to smooth lines and list estimated $R_{t}$ values while the vertical axes on the right reference gray histogram in the background providing counts of observed incidence. Shaded areas surrounding time-varying $R_{t}$ report estimated $R_{t}$ credible intervals (CIs) as shown in the inset legend on the top right. Dashed vertical lines compare the documented index case date to the earliest probable VSNJV introduction as estimated by the model. $R_{t}$ for 2015 is given in Figure 7.

VSNJV incubation rates in vectors ($\sigma_{v}$) were estimated to be a median of 0.38/day (0.32–0.46 95% CI) in 2014 and 0.37/day (0.30–0.44 95% CI) in 2015. Taking the reciprocal of the average (1/0.375=2.67) means that the extrinsic incubation period (EIP) converged on about a 3-day latency for both outbreak years (Figure 8). Although temperature may influence the EIP, the model estimated rate closely aligns with feeding rates and laboratory-determined EIP in *Culicoides* midges, which have shown infection as early as 3 days after feeding [1,53,91]. *Culicoides sonorensis* is one of the most prevalent midge species associated with livestock in the Western U.S. and is recognized as a competent VSNJV vector [92,93,94]. The intrinsic incubation period ($\sigma_{h}$) and host removal rate ($\gamma_{h}$) were consistent between years, showing an approximate 4.55-day delay during incubation within vertebrate hosts and a host removal or recovery time of approximately 9 days. The removal ($\gamma_{h}$) Credible Interval implies vertebrate hosts remained infectious for 7.69–11.11 days post-infection (Figure 8).

The vector biting rate ($\upsilon_{v}$), number of bites received by vertebrate hosts per vector ($\upsilon_{h}$), and vector mortality rate ($\mu_{v}$) closely matched the current understanding of *Culicoides* spp. physiology and behavior. The combined biting ranges (1/υ at 95% CI) for these parameters imply a vector feeding interval between 2.70 and 4.16 days (Figure 4), which aligns with laboratory observations made by Letchworth et al. [1] and temperature-dependent feeding behavior investigated by Rozo-Lopez et al. [53]. This feeding interval tracks *Culicoides* spp. gonotrophic cycles, which typically occur every three days and are blood-feeding dependent [95,96]. The natural background death rate ($\mu_{v}$) for vectors was consistent between outbreak years and exhibited a median of 0.08/day and a 95% Credible Interval of 0.06–0.09/day, which means that the median vector lifespan was 12.50 days (11.09–16.67 days 95% CI). A 10–13 day lifespan, allowing 2–3 gonotrophic cycles (bloodmeals), was previously determined for *Culicoides sonorensis* under laboratory conditions [53] and studies investigating the role of *Culicoides* spp. in the transmission of bluetongue virus [52].

VSNJV acquired by *Culicoides sonorensis* during a blood meal must first infect the vector’s midgut tissues, then bypass the midgut, salivary glands, and other organs involved in the virus transmission process before becoming infectious [40]. The vector heterogeneous competency parameter ($\rho_{v}$) included in the model was intended to quantify this and other innate mechanisms that potentially reduce the rate at which VSNJV-infected vectors become infectious. The $\rho_{v}$ parameter (Table 1) reflects the proportion of exposed vectors that ultimately become infectious. The median $\rho_{v}$ for 2014 was found to be 0.88 (0.81–0.94 95% CI) and 0.87 (0.79–0.93 95% CI) for 2015 (Figure 4). These estimates indicate that exposed vectors became infectious in about 88% of the simulated vectors. As stated from the inverse perspective, about 12% of vectors failed to transmit VSNJV following infection.

The probability of VSNJV transmission from an infectious vector to a vertebrate host assuming a successful bite ($\beta_{h}$) was well over 95% in both 2014 and 2015; by comparison, the probability of VSNJV uptake from an infectious host during a vector bite ($\beta_{v}$) was more varied over the outbreak cycle (Figure 4). Values for $\beta_{v}$ showed a 95% Credible Interval of 0.81–0.94/bite in 2014 and 0.78–0.93/bite in 2015. Comparable uncertainty has been found in model-based studies of other disease systems like African horse sickness [51] and bluetongue virus [49,54], in which high probability and certainty were found for vector-to-host transmission parameters, but host-to-vector parameters exhibited more variation.

Posterior predictive simulations demonstrated that SEIR-SEI parameterization could reproduce incidence rates and cumulative case numbers consistent with observations made during the 2014–2015 outbreak cycle. The number of observed cases each year exhibited excellent coverage of the 50% credible interval drawn from the simulations, indicating good predictive performance. Sensitivity analyses undertaken to assess parameter individual contributions showed that the total number of vectors ($N_{v}$) had the most impact on the force of infection (FOI) in hosts ($\lambda_{h}$) and insect ($\lambda_{v}$) populations (Figure 9). Increased vector numbers led to a higher FOI and earlier outbreak onset regardless of contact rates (υ) and VSNJV transmission probability (β), though contact rates and transmission rates did influence relative FOI magnitude. The models demonstrated even mixing, with no divergences, and Rhat values less than or equal to 1.0 were assessed for all parameters, indicating no pathological behavior. Model stability was maintained when prior distributions were confined to within 0.5 standard deviations. Detailed results from these evaluations and additional diagnostic information are available in the hyperlinked supporting information.

While the SEIR-SEI compartmental model presented in this study offers valuable insights into VSNJV transmission dynamics, it also has inherent limitations. One significant limitation is the model’s reliance on generalized characteristics for hosts and vectors, which may not accurately capture the heterogeneity present in real-world scenarios. Due to the generalized nature of the model, necessitated by limited observational and experimental data, it may overlook species-specific interactions and ecological nuances critical for precise outbreak predictions. Specifically, the model treats equids and bovids as equivalent vertebrate hosts and bases vector parametrization on a single vector (*Culicoides sonorensis*).

Moreover, the assumption of homogeneous mixing in compartmental models is a notable constraint. This assumption presumes an equal probability of contact among all individuals, which rarely reflects actual conditions. In reality, host and vector populations are often unevenly distributed across landscapes, and their contact processes and other interactions are influenced by spatial factors and environmental variability [70,97,98]. This spatial heterogeneity can significantly impact disease dynamics and transmission patterns.

One of the most crucial factors influencing VSNJV transmission is temperature [52,53,99]. Temperature is a significant climatic factor in the ecology of VSNJV vectors, like *Culicoides* midges, as it shapes midge seasonality, distribution, and abundance [97,99]. Temperature significantly affects vector behavior and disease transmission, influencing vector reproduction rates, feeding frequency, survival, and the virus incubation period within the vector. Higher temperatures can accelerate viral replication within the vector, leading to shorter incubation periods and more frequent transmission events. In comparison, lower temperatures may reduce vector activity and viral transmission rates [53].

To enhance the accuracy and applicability of VSNJV models, it is essential to incorporate spatial heterogeneity and time-varying, seasonal, and location-specific temperature variations. Additional model compartments and parameters are much needed to capture species-specific differences among vectors and hosts. This study especially highlights the need for expanded field sampling and enhanced laboratory capacity to improve understanding of vector biology, physiology, and ecology. By improving these factors, models will better reflect the complexities of VSNJV transmission and provide more reliable predictions for managing and mitigating outbreaks.

## 4. Conclusions

This study presents a SEIR-SEI compartmental model to analyze the transmission dynamics of the vesicular stomatitis New Jersey virus (VSNJV) during the 2014–2015 outbreak in the United States. The model integrates vertebrate hosts and insect vector populations to improve mechanistic understanding of VSNJV epidemiology and emphasizes the importance of considering heterogeneous competency and observation bias. Using Bayesian inference and Markov chain Monte Carlo (MCMC) methods, the study estimates key epidemiological parameters, providing insights into the force of infection, effective reproduction number ($R_{t}$), and other attributes of vectors and hosts.

Future work should focus on incorporating spatial heterogeneity and temperature-dependent factors to refine the model further and improve its predictive capabilities. Integrating species-specific differences among hosts and vectors will also enhance the model’s applicability to diverse ecological contexts.

## Figures and Tables

**Figure 1 viruses-16-01315-f001:**
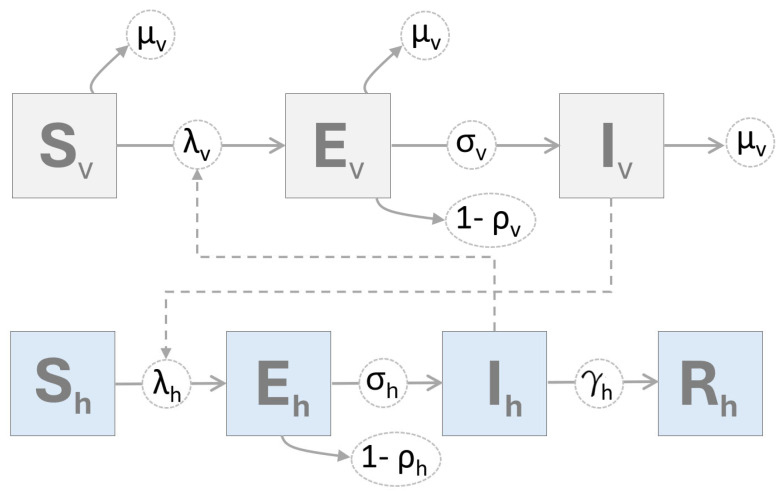
SEIR-SEI model diagram. The figure depicts a compartmental model structure with rectangles for portions of the vertebrate host population that are susceptible ($S_{h}$), exposed ($E_{h}$), infectious ($I_{h}$), and removed ($R_{h}$) from the VSNJV transmission cycle and the numbers of insect vectors that are susceptible ($S_{v}$), exposed ($E_{v}$), and infectious ($I_{v}$). Greek characters represent mechanistic parameters controlling transition rates between compartments, such as the force of infection (λ), incubation period (σ), and competency (ρ) for hosts (h) and vectors (v), as well as natural vector mortality ($\mu_{v}$) and the host removal rate ($\gamma_{h}$) as listed in Table 1.

**Figure 2 viruses-16-01315-f002:**
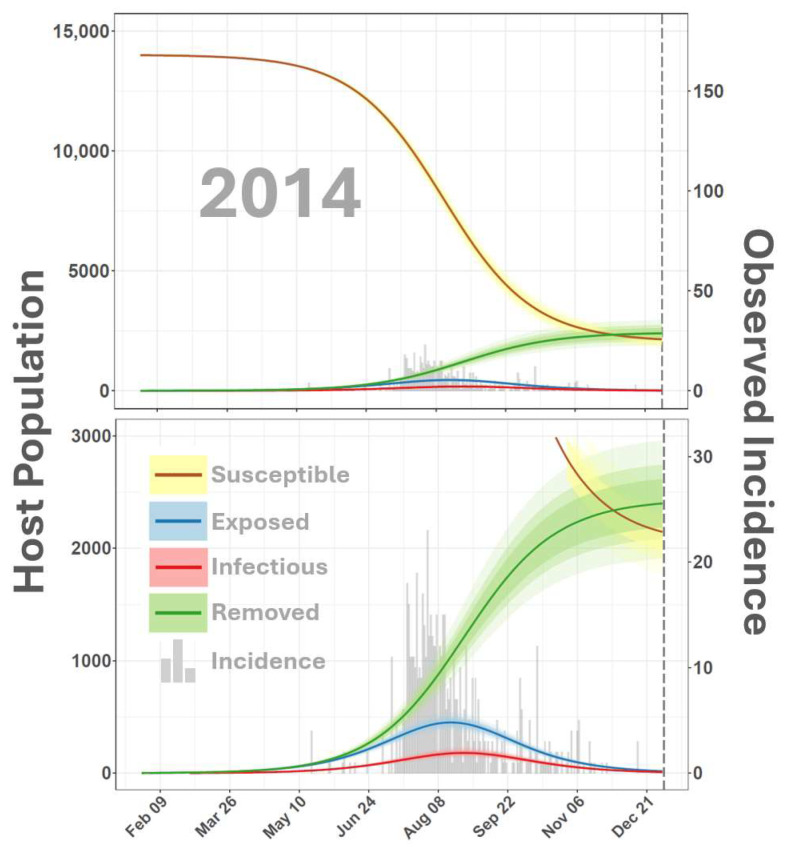
SEIR-SEI host dynamics for 2014. The figure depicts a time-varying relationship among members of the host population classified as susceptible (S), exposed (E), infectious (I), or removed (R) from the VSNJV transmission cycle. The vertical axis on the left corresponds to smooth lines and indicates host number, the secondary vertical axis on the right corresponds to gray histograms in the background and reports documented VS cases. The horizontal axis indicates the calendar date (month and day) in daily increments. Solid lines are color-coded according to the legend at the bottom left to distinguish S, E, I, and R compartments with shaded areas surrounding lines giving the 95% Credible Interval. The top panel displays the full range of simulated values, and the vertical axis in the bottom panel is truncated for detail. Model dynamics for 2015 are presented in Figure 3.

**Figure 3 viruses-16-01315-f003:**
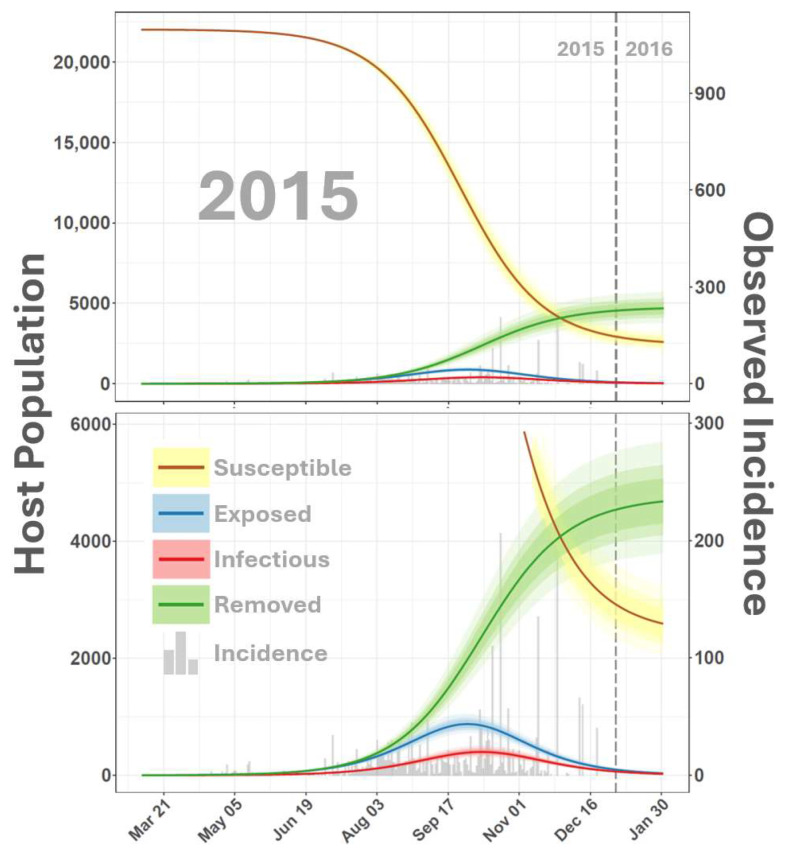
SEIR-SEI host dynamics for 2015. The figure depicts a time-varying relationship among members of the host population classified as susceptible (S), exposed (E), infectious (I), or removed (R) from the VSNJV transmission cycle. The vertical axis on the left corresponds to smooth lines and indicates the host number, the secondary vertical axis on the right corresponds to gray histograms in the background and reports documented VS cases. The horizontal axis indicates the calendar date (month and day) in daily increments. Solid lines are color-coded according to the legend at the bottom to distinguish S, E, I, and R compartments with shaded areas surrounding lines giving the 95% credible interval. The top panel displays the full range of simulated values, and the vertical axis in the bottom panel is truncated for detail. Model dynamics for 2014 are presented in Figure 2.

**Figure 4 viruses-16-01315-f004:**
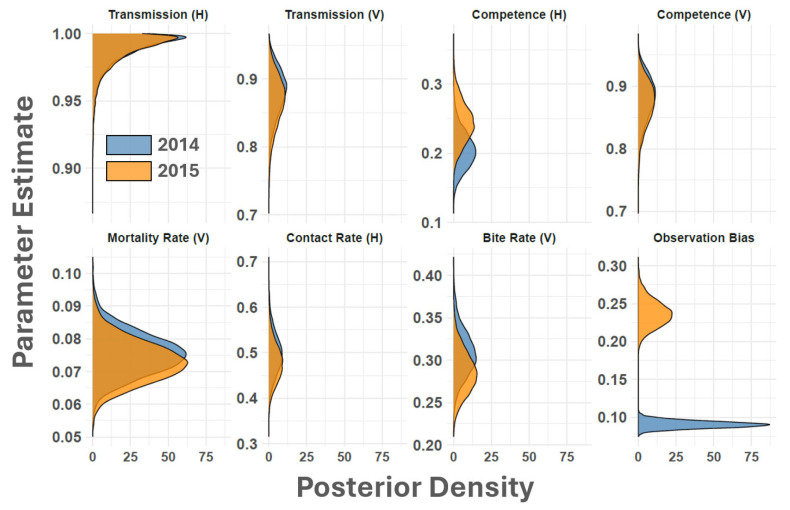
Parameter posterior densities. The figure displays estimated parameter distributions for VSNJV vectors (V) and vertebrate hosts (H) and observation bias. Axes on the left provide the parameter estimate with the horizontal axes reporting posterior density. From top left to bottom right, parameter values are given for vector-to-host transmission probability ($\beta_{h}$), host-to-vector transmission probability ($\beta_{v})$, host heterogeneous competence ($rho_{h}$), vector heterogeneous competence ($rho_{v}$), vector mortality rate ($\mu_{v}$), number of vector bites received by host ($\upsilon_{h}$), vector bite rate ($\upsilon_{v}$), and observation bias (κ). Density plots are color codes according to the legend at the bottom right to indicate the VS outbreak year.

**Figure 5 viruses-16-01315-f005:**
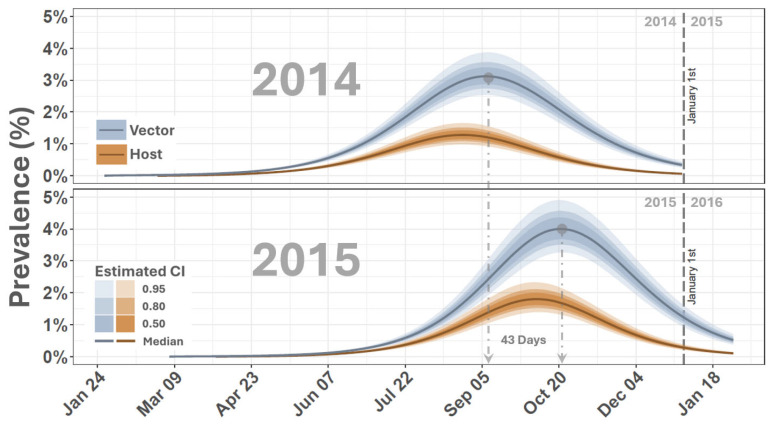
VSNJV prevalence within host and vector populations. The figure depicts the percentage of VSNJV prevalence within vector and host populations by date and outbreak year. The vertical axes list percentages, and the horizontal axis gives dates. Vector and host populations are distinguished by color as shown in legends on the left, with the solid line being the median and surrounding shaded area highlighting associated credible intervals. The top panel displays results for 2014 and the bottom panel those for 2015. Dashed vertical lines demarcate January 1st of each year for reference.

**Figure 7 viruses-16-01315-f007:**
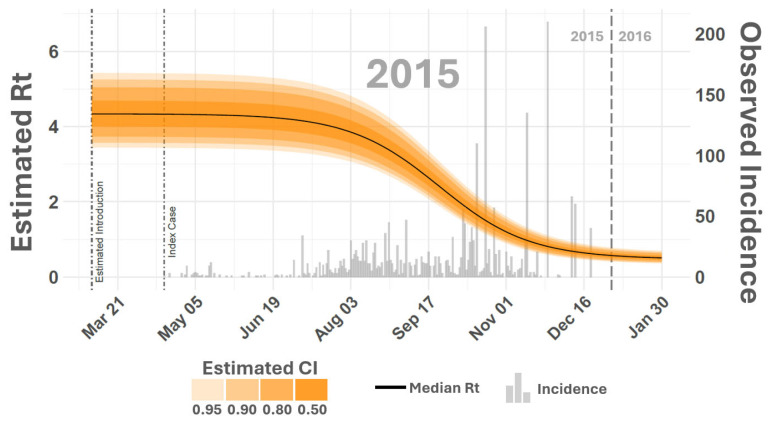
Effective reproduction number ($R_{t}$). The figure displays the estimated effective reproduction number ($R_{t}$) for 2015. The vertical axes on the left correspond to smooth lines and list estimated $R_{t}$ values while the vertical axes on the right reference gray histogram in the background providing counts of observed incidence. Shaded areas surrounding time-varying $R_{t}$ report estimated $R_{t}$ credible intervals (CIs) as shown in the inset legend on the top right. Dashed vertical lines compare documented index case dates to the earliest probable VSNJV introduction as estimated by the model. $R_{t}$ for 2014 is given in Figure 6.

**Figure 8 viruses-16-01315-f008:**
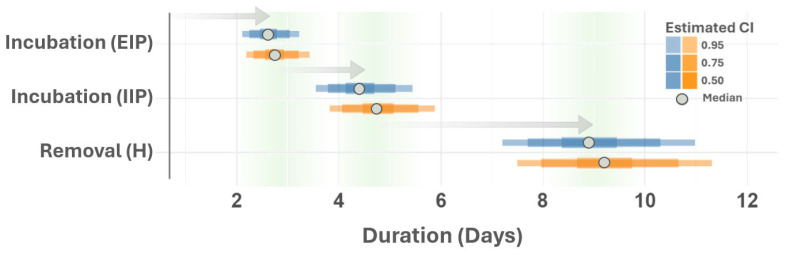
VSNJV incubation periods and host removal time. The figure depicts the estimated vector (EIP) and vertebrate host (IIP) incubation periods and the vertebrate host removal time. The vertical axis displays parameter names and the horizontal axis lists the duration periods (days). Median parameter estimator estimates are shown as a gray point with associated credible intervals (CIs) shaded according to the legend on the right. Parameters estimated for 2014 and 2015 are distinguished by color (blue and orange, respectively). Arrows are provided to highlight relative order during VSNJV transmission, with extrinsic incubation period (EIP, $\sigma_{v}$), preceding intrinsic incubation period (IIP, $\sigma_{h}$), and ultimate host removal or recovery ($\gamma_{h}$).

**Figure 9 viruses-16-01315-f009:**
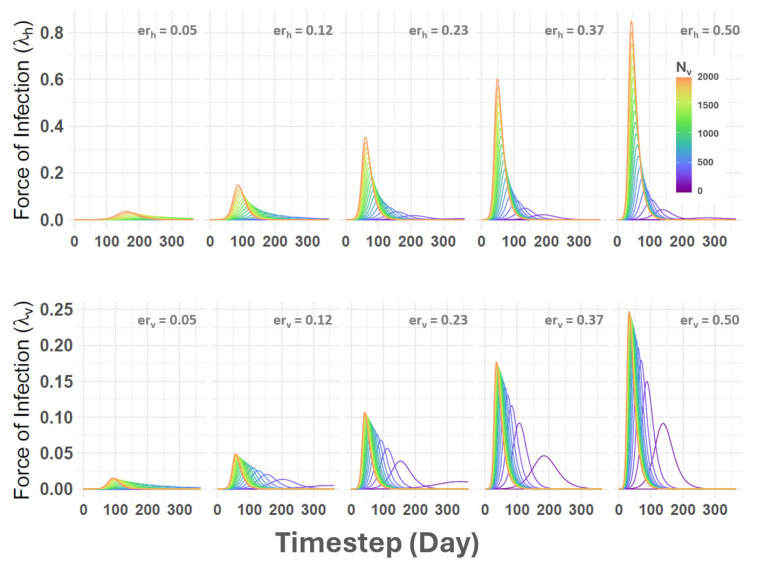
Force of infection (FOI) sensitivity to vector population and effective contact rate. The figure provides a sensitivity analysis example for the host ($\lambda_{h}$, top row) and vector ($\lambda_{v}$, bottom row) force of infection. The vertical axes on the left list FOI estimates and the horizontal axes show daily timesteps for arbitrary calendar days from VSNJV introduction (Day 0) until the end of the year (Day 365). Curved lines indicate FOI parameter distribution and are color-coded according to vector population size ($N_{v}$) as given in the inset legend in the top-right panel. Values labeled $er_{h}$ and $er_{v}$ reference effective contact rates calculated as the product of contact rates and VSNJV transmission probability for vertebrate hosts ($\upsilon_{h}*\beta_{h}$) and vectors ($\upsilon_{v}*\beta_{v}$).

**Table 1 viruses-16-01315-t001:** SEIR-SEI evaluated parameter ranges. Columns provide a brief description and the evaluated numeric range for each parameter. Parameter values determined to be optimal for the model are shown in Equation (Equation 10). Parameter biological interpretation is provided in the Results and Discussion section and parameter relationships are illustrated in Figure 1.

Parameter	Description	Evaluated Range	Reference/Notes
$N_{h}$	Total number of vertebrate hosts	3200–40,000	Data ^a^
$N_{v}$	Total number of insect vectors	1–2 × $10^{8}$	[49,50,51] ^b^
$\upsilon_{h}$	Contact rate, bites sustained by host	0.25–0.33/vector×day	[49,52,53] ^c^
$\upsilon_{v}$	Contact rate, vector biting rate	0.25–0.33/day	[49,52,53]
$\beta_{h}$	Vector-to-host transmission probability	0.05–0.50/bite	[49,51,54,55]
$\beta_{v}$	Host-to-vector transmission probability	0.75–1.00/bite	[49,51,54,55]
$\sigma_{h}$	Intrinsic Incubation Period (IIP)	0.05–0.50/day	[41,53,56]
$\sigma_{v}$	Extrinsic Incubation Period (EIP)	0.10–0.60/day	[51,53,54,55]
$\gamma_{h}$	Host removal rate (recovery or quarantine)	0.05–0.20/day	[54,55]
$\rho_{h}$	Host heterogeneous competency	0.01–1.00 (∝/day)	unlimited ^d^
$\rho_{v}$	Vector heterogeneous competency	0.01–1.00 (∝/day)	unlimited ^d^
$\mu_{v}$	Vector background mortality rate	0.05–0.50/day	[49,52,54,55]
κ	Observation bias (proportion observed)	0.01–1.00 (∝/day)	unlimited ^e^

^a^ Minimum value reflects the total reported cases, maximum is the sum of all incidence and susceptible animals rounded up to the nearest 10,000 (see, Section 2.1). ^b^ Maximum range reflects 5000 vectors per vertebrate host at $N_{h}$ = 40,000. ^c^ Bites received by the host, per biting vector, per day. ^d^ Proportion per day. Range inclusive of all possibilities however, several sources describe rates and proportions, see [6,10,41,57]. ^e^ Proportion per day. Range inclusive of all possibilities. The value is estimated during model fitting (Equation (Equation 10)) through the reconciliation of observed case numbers with those simulated based on mechanistic parameters.

## Data Availability

Code and data to conduct the analyses presented in this manuscript have been archived on the Open Science Framework (DOI 10.17605/OSF.IO/RHZU8, https://osf.io/vqgxs/ (accessed on 16 August 2024)). A tutorial to construct the models is provided at https://geoepi.github.io/seir-vector/overview (accessed on 16 August 2024), and an interactive dashboard to evaluate model parameters is also provided (https://geoepi.shinyapps.io/seir-vector/ (accessed on 16 August 2024)).

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
