# Peer review of "Modeling the 2014–2015 Vesicular Stomatitis Outbreak in the United States Using an SEIR-SEI Approach"

_viruses, 2024, doi:10.3390/v16081315_

Round 1

Reviewer 1 Report

Comments and Suggestions for Authors

see the attached file.

Author Response

REVIEWER 1 

COMMENT 1: The authors have proposed an SEIR-SEI model to investigate the Vesicular stomatitis disease transmission. Bayesian inference and MCMC method have been used to estimate the parameters. However, the obtained results obtained in this paper are common and not sound enough. Thus, I do not think the manuscript can not be accepted in current edition.

RESPONSE 1: We thank the reviewer for the time spent evaluating the manuscript.  Because the reviewer provided no specific suggestions or critiques and did not indicate any areas of concern, no revisions have been made pursuant to this comment.  Furthermore, we note that in completing the “Review Report Form” that accompanies this comment, the Reviewer affirmed that the manuscript includes sufficient background and references, has an appropriate research design, is clearly presented, and that its conclusions are supported by the study results. 

Additionally, the use of the double negative in the final sentence of Comment 1 suggests to the authors that the reviewer views the manuscript as appropriate for publication in its current form.

Reviewer 2 Report

Comments and Suggestions for Authors

The manuscript by Humphreys et al. presents an interesting SEIR-SEI approach to enhance VS dynamics. The model appears robust and reliable, incorporating a variety of features, parameters, and specifications. However, the manuscript lacks details on the incorporation of stochastic processes to account for random events and uncertainties. It would be beneficial for the authors to provide more information and clear explanations on the stochastic elements included in the model specifications. Additionally, I would like to ask whether the authors have employed efficient algorithms for solving the differential equations, particularly for large populations or complex models, and if so, which ones. Furthermore, it would be advantageous to include an additional model validation against independent datasets that were not used in the model calibration. Regarding the interactive dashboard, it would be very useful if the authors could enhance its functionality by allowing users to upload their own datasets for custom simulations and providing options to download simulation results in various formats (CSV, Excel, PDF).

Author Response

REVIEWER 2 

COMMENT 1: The manuscript by Humphreys et al. presents an interesting SEIR-SEI approach to enhance VS dynamics. The model appears robust and reliable, incorporating a variety of features, parameters, and specifications.

RESPONSE 1: The authors thank Reviewer 2 for the time spent evaluating the manuscript and appreciate the conclusion that the model appears robust and reliable.

COMMENT 2: However, the manuscript lacks details on the incorporation of stochastic processes to account for random events and uncertainties. It would be beneficial for the authors to provide more information and clear explanations on the stochastic elements included in the model specifications.

RESPONSE 2: We thank the reviewer for identifying the need for clarification regarding deterministic and stochastic models generally and how random variation is incorporated into our modeling framework specifically.  The manuscript has been revised to address the reviewer's concerns.

As an explanation here, population-level transmission models divide the overall population into distinct groups known as compartments, where individuals in each group share the same disease status (e.g. susceptible, exposed, infectious, removed). The transitions between these compartments can be modeled deterministically or stochastically. Deterministic models utilize ordinary differential equations (ODEs) to describe the rates of change between compartments and produce a consistent outcome for a given set of parameters and initial conditions by numerically solving the ODEs. By comparison, stochastic models incorporate random variation in the transitions between compartments. Stochastic models are typically constructed using Stochastic Differential Equations (SDEs), discrete event simulation, or agent-based models.  

The VSV model in the paper applies ODEs and is deterministic, not stochastic.  However, an area of confusion may exist in that, due to imperfect surveillance data and imperfect knowledge of the VSV  transmission mechanisms, we chose to introduce some randomness in our model by including a Negative Binomial sampling distribution, which does add random noise and variation to estimates from the ODEs but not to the compartment transitions. 

The manuscript has been revised to include this clarification, please see LINES 227-235 of the revised manuscript.

COMMENT 3: Additionally, I would like to ask whether the authors have employed efficient algorithms for solving the differential equations, particularly for large populations or complex models, and if so, which ones. Furthermore, it would be advantageous to include an additional model validation against independent datasets that were not used in the model calibration.

RESPONSE 3: Thank you for reminding us to specify the exact algorithm used.  Although the original manuscript described the use of the Stan language, it neglected to mention the  Runge‐Kutta (RK45) algorithm, which is available in Stan and used to solve the ODEs.  The manuscript has been revised to clarify the use of Runge‐Kutta solver and includes citations attesting to its validity and reliability.  Please see LINES 254-255 of the revised text.   

The authors fully agree with the reviewer's inclination to suggest additional model assessment using independent data sets; however, no other surveillance data are available for the 2014-15 US outbreak cycle, it would not be appropriate to calibrate the current model with a subset of what data are available for 2014-15, and use of data from other diseases or different VS outbreak cycles would exceed the current manuscript’s objective and purpose, which is to assess, analyze, and describe 2014-15 outbreak dynamics.

While we appreciate the reviewer’s suggestion, the current manuscript focuses on analyzing the 2014-2015 outbreak cycle. We intend to develop a more general model capable of predicting and forecasting VS outbreaks across various contexts and times, but this broader objective is beyond the scope of the current project and remains a future research goal.

Please see the revised manuscript text at LINES 446-458 and LINES 468-475, emphasizing the current study’s scope and limitations.

COMMENT 4: Regarding the interactive dashboard, it would be very useful if the authors could enhance its functionality by allowing users to upload their own datasets for custom simulations and providing options to download simulation results in various formats (CSV, Excel, PDF).

RESPONSE 4: As with the reviewer’s suggestion to develop a more general-purpose VS model capable of estimating outbreaks in varied contexts and periods, the authors fully agree that a software application, tool, or interactive dashboard to make the presented framework more accessible to others is needed and would be valuable.  Although the authors have discussed expanding the current project to these suggested areas, those goals remain future objectives and are beyond the scope of the current manuscript,  which is to assess, analyze, and describe 2014-15 outbreak dynamics.

We thank the reviewer for their valuable comments and suggestions. We believe the reviewer’s contributions have substantively improved the manuscript and will help motivate expanded research in VS model-based analysis.

Reviewer 3 Report

Comments and Suggestions for Authors

There are considerable challenges in jointly modelling viruses such as vesicular stomatitis and their vectors, especially in this case where there are multiple ecologically different vectors and multiple virus types.  There are not many data sets to use in refining such a model, and they have used one such data set to good effect. This paper does an excellent job of progressing the development of such models by making some simplifications (such as a single vector).  They describe the simplifications in the Discussion and provide a balanced assessment of where they have reached in modelling this rather unusual disease.  Seen within this context, I consider this a valuable paper and support its publication. 

Author Response

REVIEWER 3

 COMMENT 1: There are considerable challenges in jointly modelling viruses such as vesicular stomatitis and their vectors, especially in this case where there are multiple ecologically different vectors and multiple virus types.  There are not many data sets to use in refining such a model, and they have used one such data set to good effect. This paper does an excellent job of progressing the development of such models by making some simplifications (such as a single vector).  They describe the simplifications in the Discussion and provide a balanced assessment of where they have reached in modelling this rather unusual disease.  Seen within this context, I consider this a valuable paper and support its publication.

RESPONSE 1: We thank the reviewer for evaluating the manuscript, providing supportive comments, and recognizing the inherent challenges in capturing a complex and dynamic disease in a simple modeling framework.

Round 2

Reviewer 1 Report

Comments and Suggestions for Authors

The paper can be accepted in current edtion.